# Joint Toxicity of a Multi-Heavy Metal Mixture and Chemoprevention in Sprague Dawley Rats

**DOI:** 10.3390/ijerph17041451

**Published:** 2020-02-24

**Authors:** Yafei Wang, Yuqing Tang, Zhou Li, Qihang Hua, Li Wang, Xin Song, Baobo Zou, Min Ding, Jinshun Zhao, Chunlan Tang

**Affiliations:** 1Department of Preventative Medicine, Zhejiang Key Laboratory of Pathophysiology, School of Medicine, Ningbo University, Ningbo 315211, Zhejiang, China; wangyafei926@163.com (Y.W.); Yuqing_tang897@163.com (Y.T.); Lizhou0601@163.com (Z.L.); huaqihang@nbu.edu.cn (Q.H.); songxin3c@163.com (X.S.); zoubaobo@nbu.edu.cn (B.Z.); 2Ningbo Municipal Center for Disease Control and Prevention, Ningbo 315010, Zhejiang, China; ningbowl@163.com; 3Toxicology and Molecular Biology Branch, Health Effects Laboratory Division, National Institute for Occupational Safety and Health, Morgantown, WV 26505, USA; mid5@cdc.gov

**Keywords:** multi-heavy metal mixture, joint toxicity, accumulation, chemoprevention, antagonism, Sprague Dawley rats

## Abstract

To explore the joint toxicity and bio-accumulation of multi-heavy metals and potential chemoprevention strategies, Male Sprague Dawley (SD) rats (*n* = 30) were treated orally once a week for six months with 500mg/kg•bw of eight heavy metals which were commonly identified in aquatic products in the Ningbo area including chromium, manganese, nickel, copper, zinc, cadmium, mercury, and lead. At the same time, 200mg/kg•bw of epigallocatechin-3-gallate (EGCG), trisodium citrate dihydrate (TCD) or glutathione (GSH) were administered to evaluate their antagonistic effects against adverse effects of multi-heavy metal mixture. The Morris water maze test was used to evaluate spatial learning and memory in the treated rats. Then the rats were anesthetized by pentobarbital sodium (40 mg/kg•bw) to obtain blood samples for biochemical analysis and organs (heart, liver, spleen, lungs, kidneys, brain, testis) to be conducted for biopsy and organ coefficients. Inductively coupled plasma mass spectrometer (ICP-MS) was used to analyze the concentrations of heavy metals. Results indicated that six months of exposure to a multi-heavy metal mixture under this experimental dosage resulted in accumulation in organs and adverse effects on the blood, reproductive system, and liver function. EGCG, TCD or GSH all showed certain chemoprevention effects against the joint toxicity induced by the multi-heavy metal mixture and indicated alleviation and the potential mechanism that also included the promotion of excretion of metals to which animals were exposed.

.

## 1. Introduction

Heavy metals are widely used in industries. Lead (Pb) and mercury (Hg) are used predominantly in batteries, glass, building, crude oil, and automobile manufacturing, while cadmium (Cd) and nickel (Ni) are essential for electroplating, dyes, and plastics [1,2]. As industrialization increases around the globe, so too does the emission of heavy metals into the environment. The accumulation of these emissions in the air, soil, plants, water, and sewage sludge has overwhelmed the natural capacity in many ecosystems, resulting in the potential for humans to be exposed to heavy metals [3,4]. Because heavy metals are not easily biodegradable and can persist in the environment, heavy metals may exert long-lasting and permanent adverse effects once entering into the human body through contaminated air, water or food [5].

Cd is recognized as an environmental carcinogenic pollutant with a biological half-life of more than 20 years. Once it enters into human bodies, Cd easily accumulates in the tissue and binds itself with thiols of metallothionein, which can disturb the activity of antioxidant enzymes, breach the normal structure of DNA and finally lead to teratogenicity and carcinogenicity [3,6,7]. Hg is also a highly toxic and bio-accumulative heavy metal. Hg is highly soluble, which allows it to liberally penetrate the pulmonary alveoli, blood-brain barrier and blood-testis barrier, causing neurotoxicity, reproductive toxicity, renal toxicity, and liver toxicity, among its detrimental effects [8]. Pb is a traditional environmental heavy metal that has been used in industries for thousands of years including for the manufacture of plastic, paints, ceramics, glass, water pipes, and insecticides [9]. Pb can induce neurotoxicity, hepatotoxicity, and nephropathy [10,11]. As necessary microelements for physiological activities, chromium (Cr), manganese (Mn), Ni, copper (Cu), and zinc (Zn) play a significant part in normal development and organisms’ homeostasis maintenance. However, high-level exposure to these elements may also induce adverse health effects [12,13,14,15,16]. Although the toxic molecular mechanisms of different heavy metals vary, they also have some similarities. When entering into organisms, they can disturb intracellular homeostasis and inhibit proteinase activity, leading to excessive generation of reactive oxygen species (ROS), DNA damage, and carcinogenicity [17,18,19,20]. These similarities contribute to searching for effective antioxidative chemicals for preventing multi-heavy metal mixture induced toxicities. 

Epigallocatechin-3-gallate (EGCG), a polyphenol extracted from green tea, reportedly possesses strong antioxidant and anticancer characteristics [21,22]. Active hydroxyl in EGCG can provide hydrogen atoms for binding with excessive free radicals in vivo, participate in the coenzyme II electron transfer chain to disrupt free radical reaction, and therefore alleviate the damage of ROS to biological macromolecules [23]. In addition, EGCG is a metal chelating agent. Phenolic OH groups, the component of EGCG, react with metal ions as a ligand and create cyclic chelate, which is very slightly soluble. Therefore, EGCG has a potential protective effect against heavy metal-induced toxicity [24]. 

Citric acid is a weak organic acid existing in many citrus fruits, such as lemon, grapefruit, and orange. As a crucial component of tricarboxylic acid, citric acid can be found in almost all organism tissues and plays an important role in antioxidant and cellular immunity [25,26,27]. It can also alleviate oxidative stress and attenuate inflammation and DNA damage [28]. Therefore, citric acid may be a potentially effective antidote to inhibit the joint toxicity induced by multiple heavy metals. 

Glutathione (GSH), a three-peptide thiol, widely exists in animals and plants like tomatoes, yeast, wheat germ, and animal liver. Researchers have highlighted its varied functions, including as an antioxidant, chelating agent, and signal transductor [29]. The diverse functions of GSH originate from the sulfhydryl group in cysteine, enabling GSH to chelate metals and participate in redox cycling. In the presence of glutathione peroxidase (GSSG), sulfhydryl compounds reduce intracellular hydrogen peroxide (H2O2) into H2O, and themselves are oxidized to GSSG. Under the condition of GSH reductase in liver and erythrocyte, GSSG receives H and can be reduced to GSH. In this process, the damage of free radicals caused by heavy metals can also be reduced. In addition, sulfhydryl can combine with heavy-metal ions to form soluble chelation and increase the chelation discharging from organisms, thus reducing toxic effects and accumulation of heavy metals [29,30,31]. Consequently, GSH seems to be a potential antagonist against heavy metals. 

Toxicity and accumulation characteristics of single heavy metal have been well documented [32,33]. However, the human body is normally exposed to multiple heavy metals simultaneously. Ningbo, as a coastal city of China, is abundant in aquatic products. Our previous study has found that Cr, Mn, Ni, Cu, Zn, Cd, Hg and Pb were the most common heavy metal pollutants in aquatic products [34]. The objective of this study was to investigate the joint toxicity following chronic exposure to a multi-heavy metal mixture through in vivo experimentation. For exploring chemoprevention agents, three chemical antagonists—EGCG, trisodium citrate dihydrate (TCD) and GSH—were evaluated concurrently.

## 2. Materials and Methods

### 2.1. Materials 

K_2_Cr_2_O_7_, MnCl_2_•4H_2_O, NiCl_2_•6H_2_O, CuSO_4_•5H_2_O, ZnSO_4_•7H_2_O, CdCl_2_•2.5H_2_O, CH_3_ClHg, CH_3_COO)_2_Pb•3H_2_O, TCD (>99%), 0.9% sterile normal saline, 40% formaldehyde solution were purchased from Sinopharm Chemical Reagent Co., Ltd (Beijing, China). CH_3_ClHg (analytical grade, ≥99.0%) was purchased from Dr. Ehrenstorfer GmbH (Augsburg, Germany). EGCG (90%) was purchased from Hangzhou Yibeijia Tea Co., Ltd (Hangzhou, China). GSH (98%) was purchased from Solarbio Life Sciences Co., Ltd. (Beijing, China). Nitric acid (HNO_3_, 68%, trace metal, UP grade) was purchased from Suzhou Crystal Clear Chemical Co., Ltd. (Suzhou, China). Male Sprague Dawley (SD) rats weighing 150–180 g were purchased from the Experimental Animal Center of Zhejiang Province. Inductively coupled plasma mass spectrometer (ICP-MS) (NexION™ 300D) was produced by PerkinElmer (USA).

### 2.2. Methods

#### 2.2.1. Animal Experiment

All animals were housed in a temperature-controlled animal facility (22 ± 2 °C) with a 12-hour light/dark cycle (lights on from 7:00 AM to 7:00 PM). All rats had access to water and fodder ad libitum and received humane care in compliance with the Principles of Approval of the Experimental Animal Ethics Committee of Ningbo University (Approval No.: AEWC-2017-23; Date: 2017.01.10).

30 SD rats were randomly divided into five groups, 6 in each group (control, Mixture, Mix + EGCG, Mix + TCD, Mix + GSH). Mixture represented the group treated with the mixture containing eight heavy metals at a total exposure dosage of 500 mg/kg•bw once per week through intragastric administration. The administration dosages for EGCG, TCD, and GSH were all 200 mg/kg, following the administration of eight heavy metals once per week and the control group received the same volume of ultrapure water. This research lasted for six months. All the animals survived at last.

#### 2.2.2. Preparation of Multi-Heavy Metal Mixture 

The multi-heavy metal mixture, which included Cr, Mn, Ni, Cu, Zn, Cd, Hg, and Pb, was based on the estimated intake proportions of daily consumption of each metal through aquatic products (Table 1) in the Ningbo area according to our previous research [34]. To prevent precipitation, each heavy-metal compound was separately dissolved as a stock solution. Before use, the heavy-metal compound stock solution was mixed and diluted to the working concentration with ultrapure water. The final concentration contained the sum of eight heavy metal compounds. 

The Morris water maze test consisted of a black round pool with a diameter of 180 cm and depth of 50 cm and was divided into four quadrants (I, II, III, IV) with different specific visible references, which were fixed during the experiment. A 10-cm-diameter and 25-cm-height escape platform was located in quadrant I and remained unchanged throughout the test. The water was dyed black and opaque by carbon ink, and the water temperature was maintained at 22~24 °C. A camera system was installed above the maze and connected to the computer to automatically analyze the swimming tracks of rats. The lab was kept quiet and dark throughout the Morris water maze test.

The water maze experiment included two parts: a place navigation test and a spatial probe test. The day prior to the formal experiment, all rats were acclimated to the water and dark environment and swam freely for 2 min. During the following four days, each rat was trained for four trials per day (60 s/trial), starting from the platform quadrant, then quadrants II, III, IV in a row with an interval of 1h. The spatial probe test was performed on the final day. The platform was removed out of the pool and rats were put into the pool from quadrant III. Times of crossing the platform were recorded in 60 s. The data were handled with Any-maze software (EthoVision XT7.0, Noldus Information Technology b.v., Wageningen, The Netherlands).

#### 2.2.3. Organ Weight and Organ/Body Weight Coefficients 

After the Morris water maze test, 30 rats were weighed and sacrificed. Organs, including the brain, heart, liver, spleen, lung, kidney, and testicle were obtained immediately. Parts of the fresh tissues were obtained for histopathological examination in formalin for at least 48 h and the rest were stored at −80 °C for ICP-MS analysis. Organ weight and organ/body weight coefficients were calculated subsequently.

#### 2.2.4. Hematological and Biochemical Analysis

After the animal was anesthetized by pentobarbital sodium (40 mg/kg•bw), 1 mL of blood was collected in test tubes containing 20 mg/mL EDTAK2 anticoagulant for complete blood count (CBC) from the apex of the heart. Another 4 mL was placed into tubes containing heparin sodium. The sera were separated by centrifugation at 4 °C, 3000 rpm, 15 min immediately. Hematological and biochemical analysis was conducted at the time when we got the samples.

#### 2.2.5. Histopathological Examination

The formalin-fixed tissues were stored at 4 °C until examination. Tissues were processed using standard histology laboratory techniques. They were first fixed in formalin, then embedded in paraffin wax and cut into 3–4 μm sections for hematoxylin and eosin (H&E) staining.

#### 2.2.6. Determination of Heavy Metals in Serum and Organs by ICP-MS

Serum samples were diluted 20 × with 1%HNO_3_ solution and then were used for detecting eight heavy-metal elements by ICP-MS.

The contents of eight heavy metals in tissues including heart, liver, spleen, lung, kidney, brain, and testicle were analyzed by ICP-MS after microwave-assisted acid digestion. A precise 0.5 g-sample of organic tissue was pre-digested with 5.5 mL of 68%HNO_3_ in PTFE vessel at 110 °C for 30 min and then transferred into a microwave digestion system (MARS 6 classic, CEM, USA) for further mineralization. Three blank samples were tested in the same way. Digestion conditions for MARS 6 classic were applied as follows: up to 180 °C for 25 min and then constant for 20 min; a cooling stage (30 min) carried out to 80 °C. After cooling to room temperature, digestion samples were diluted with ultrapure water. These samples were used for the final heavy-metal element analysis, performed with an ICP-MS equipped with a concentric Nebulizer, a quartz torch with a quartz injector tube, and a cyclonic spray chamber.

### 2.3. Statistics

Statistical analysis was performed using one-way or two-way ANOVA followed by LSD-t test to compare multiple experimental treatments by SPSS 16.0 software package (SPSS Inc., Chicago, IL, USA). Data were presented as means ± standard deviation (SD). A *p*-value less than or equal to 0.05 was considered statistically significant.

## 3. Results

### 3.1. Morris Water Maze Test

As shown in Figure 1A, escape latency results showed no significant difference when compared with the control group. The analysis for platform crossing times, as shown in Figure 1B, showed that times in the Mixture and Mix + EGCG groups were lower than times in the control group, although no significant difference was found. However, platform crossing times in the Mix+GSH group was higher than in the control and Mixture groups.

### 3.2. Organ Weight and Organ/Body Weight Coefficients

Organ weight, organ/body weight coefficients are shown in Table 2 and Table 3. The absolute weights, organ/body weight coefficients of the heart, liver, spleen, lung, kidney, and brain showed no significant differences among all groups. However, the absolute weight of the testicle was lower in the groups treated with the heavy metal mixture compared to the control group (*p* < 0.05). Organ/bodyweight coefficients of testicle in Mix + EGCG, Mix + TCD, and Mix + GSH groups were lower when compared with the control group (*p* < 0.05). The coefficient of the lung in the Mixture group was higher than the other four groups (*p* < 0.05).

### 3.3. Hematological and Biochemical Analysis

The hematological effects of different treatments are shown in Table 4. No significant difference was observed in white blood cell counts. Red cell distribution width (RDW-CV) of the Mixture group was significantly higher than the control, Mix + EGCG and Mix + GSH groups (*p* < 0.05). Mean platelet volume (MPV) in all the heavy-metal treatment groups was higher than in the control group (*p* < 0.05).

Mix, Mixture; WBC, whole blood count; NE%, neutrophil%; LY%, lymphocyte%; MO%, monocyte%, EO%, eosinophil%; RBC, red blood cells; HGB, hemoglobin; HCT, hematocrit; MCV, average erythrocyte volume; MCH, mean corpuscular volume; MCHC, mean corpuscular hemoglobin concentration; RDW-CV, red cell distribution width; PLT, platelet; PDW-CV, platelet distribution width%; MPV, mean platelet volume; PCT, plateletcrit. 

A biochemical analysis of the serum is presented in Table 5. The total bilirubin (TBIL) and indirect bilirubin (IBIL) of the Mixture group were significantly higher than the control group (*p* < 0.05) and the addition of EGCG, TCD or GSH could significantly alleviate these effects. Alkaline phosphatase (ALP) in the Mixture and Mix+EGCG were significantly higher than the control group (*p* < 0.05).

Mix, Mixture; TG, triglycerides; CHOL, total cholesterol; TBIL, total bilirubin; DBIL, direct bilirubin; IBIL, indirect bilirubin; TP, total protein; ALB, albumin; GLOB, globulin; A/G, albumin/globulin ratio; ALT, alanine aminotransferase; AST, aspartate transaminase; ALP, alkaline phosphatase; UREA, urea nitrogen; CREA, creatinine; UR/Cr, urea/creatinine ratio; URCA, uric acid.

### 3.4. Histopathological Examination 

Pathological effects induced by multiple heavy metals in tissues were evaluated by hematoxylin and eosin (HandE) staining. The liver and testicle tissues in the mixture-treated group showed inflammatory cell infiltration and histopathological changes. The addition of EGCG, TCD or GSH attenuated these adverse effects significantly (Figure 2).

### 3.5. ICP-MS Results

The contents of eight heavy metals in sera and organs are shown in Table 6 and Figure 3, Figure 4, Figure 5 and Figure 6. Isotopes ^52^Cr, ^55^Mn, ^60^Ni, ^63^Cu, ^66^Zn, ^111^Cd, ^202^Hg, and ^208^Pb were used to detect and analyze eight heavy metals. The eight heavy metals were distributed differently in sera and organs.

## 4. Discussion

A large body of research has investigated the toxicity of one or two kinds of heavy metals, but few articles have studied the joint toxicities of multi-heavy metals. However, people are normally exposed to multiple heavy metals simultaneously either through contaminated water, air or food. Multi-heavy metals exposure may result in a series of complicated consequences including accumulation followed by tissue or organ damage. This study tried to evaluate the joint toxicities and to search for chemoprevention strategies through simulating the actual dietary intake ratio of eight common heavy metals known to contaminate aquatic products in the Ningbo area.

The Morris water maze test is a useful approach to check neurological disorder after hazardous factor exposure. After six months exposure, no obvious neurotoxicity effects were observed through the Morris water maze test in the groups treated with heavy metals, although a small amount of Cd and Hg accumulation could be detected in the rat brains by ICP-MS. Platform crossing times, which indicated the animals’ memory function, were relatively lower in the mixture treated group than the other groups but there was no significant difference compared to the control.

Exposure to the multi-heavy metal mixture resulted in a significant increase of Cr, Ni, and Hg contents accompanied by an elevated level of RDW-CV and MPV in serum. Elevated RDW-CV suggested that the inhomogeneous size of erythrocytes occurred followed by hematopoietic dysfunction. Elevated MPV is considered as a common index of platelet activity positively correlated with thromboxane A2 secretion, which represents an independent risk factor for myocardial infarction and stroke [35]. To a certain degree, additions of EGCG, GSH or TCD could decrease the content levels of Cr, Ni, and Hg in serum, and increase the level of RDW-CV in the heavy metal mixture treated groups. These results suggest that EGCG, GSH or TCD may be helpful to promote the excretion of Cr, Ni or Hg and then to attenuate the blood system injury.

ICP-MS results indicated that Zn, Hg, and Cd could accumulate in liver tissues. Serum biochemical indices showed that TBIL, IBIL, and ALP all increased in the animals treated with the multi-heavy metal mixture. These results were consistent with the histopathological findings of dilated central veins and infiltrated lymphocyte. Our results showed that the addition of EGCG, TCD or GSH prevented induced liver toxicities in the animals treated with the multi-heavy metal mixture.

In the testis, contents of Cr, Cu, Cd, and Hg increased significantly and a reduction of testicle weight was observed after six months exposure to the multi-heavy metal mixture. Organ weight is considered the most sensitive indicator of toxicity. It often precedes the morphological changes, organ dysfunction or atrophy. Histopathological examination showed a decreased spermatogenic cell number and glial hyperplasia. Previous studies reported that long-term exposure of Cd or Hg could reduce the activity of antioxidant enzymes, aggravate oxidative damage, and eventually lead to testicular weight reduction and sperm deformity [2,36]. In vitro experiments suggested that Cr exposure could damage the reproductive system, and treatment with hexavalent chromium resulted in reduced sperm count and loss of motility in rats [37]. These findings are all consistent with our results. 200 mg/kg•bw of EGCG, TCD, or GSH could apparently alleviate such pathological damage, but this treatment would neither decrease the accumulation of Cr, Cu, Cd, and Hg in the testicle nor reverse the decreased testicular weight caused by the multi-heavy metal mixture. Because these chemical antagonists could not effectively reduce the accumulation of heavy metals in the testicle, irreversible damage of the reproductive system might eventually occur if the exposure continues.

Renal histopathology and function perimeters such as UREA, CREA, and URCA remained in the normal range, while the levels of Cu, Zn, Hg, Cd, and Pb in the kidneys were observed to increase. These results indicated that the kidneys might still be in a compensatory stage. Similar results were observed in the heart, lungs, and spleen; no significant adverse histopathological or biochemical changes were observed, but heavy metal accumulation, especially Hg and Cd, occurred in different degrees in these organs. 

The contents of some heavy metals such as Cu and Zn showed a downward trend compared to the control. In the present study, this reduction was observed in the spleen and brain. Erdem O et al. reported that superoxide dismutase activity in liver, heart, and kidney decreased and glutathione peroxidase activity significantly increased accompanied by reduced Cu and Zn after SD rats were given 15mg/L of cadmium chloride orally for 8 weeks. Their results suggest that a high concentration of Cd might affect the concentration of biological elements, such as Cu and Zn, in vivo [38,39]. In addition, a suitable concentration of Zn had an antagonistic effect on the toxicity of Cd. Saibu Y et al. also proved that when treated with Zn and Cd, the Cd content in gills increased, but the content of Zn decreased, suggesting antagonistic effects between Zn and Cd [40]. This may be attributed to competition between Zn and Cd when combined with metallothionein [39]. 

Antagonistic effects also existed between Pb and Zn. Zn is an active component of many proteins, amino acids, and metalloenzymes in the organisms, and plays a variety of physiological functions, which could be replaced by Pb causing abnormal homeostasis and antagonizing the physiological effects of Zn [41]. Ni in the liver and brain also decreased, but the underlying mechanism was still unclear. As for Mn, literature data suggest that, under natural physiological conditions, Mn accumulates extensively in human bones [42]. Over repeated administration, Mn selectively accumulates in the brain, leading to neurodegenerative damage [43]. In our experiment, we performed 45.609 mg/kg•bw MnCl_2_•4H_2_O once a week for six months, but no obvious accumulation was observed in the serum or organs and no neurotoxicity was detected. Mn in bones was not explored. Further research should be conducted on the metabolic mechanism of Mn, both alone and with multi-heavy metals. Based on our combined study results, we speculate that antagonistic effects among these eight heavy metal elements might occur.

Essential elements such as Cr, Mn, Ni, Cu, and Zn normally accumulate only in one or a few organs; by contrast, detrimental heavy metals like Cd and Hg could deposit in almost all organs. The reason for this phenomenon may be that organisms possess the capability to mediate indispensable trace elements effectively and to regulate their intake and excretion to maintain internal homeostasis. Researchers have identified that Cu binding proteins like CTR1 were a necessary mechanism for Cu uptake and transport [44], and ATP7B in hepatocytes was vital to help mammals excrete excess Cu [45,46]. In human bodies, proteins like ZIP4 and ZnT1 mediate dietary Zn uptake and distribution to the organism [47]. Regulation of Zn excretion is mainly through the gastrointestinal tract. Zn is excreted in the feces through food consumption, and fecal Zn also derives from pancreatic and biliary secretion into the intestine [48]. As a consequence, to maintain normal physiological functions, organisms have developed an effective biological mechanism to consume and excrete obligatory trace elements, but they lack a feasible approach to eliminate detrimental heavy metals like Cd, Hg or Pb.

## 5. Conclusions

In conclusion, chronic multi-heavy metal administration in rats resulted in different degrees of accumulation in various tissues or organs: Hg and Cd accumulated in serum, heart, liver, spleen, lungs, kidneys, brain, and testis; Cr in serum, testis, and spleen; Ni in serum and lung; Zn in liver and kidneys; Cu in kidney and testis; Pb in kidneys and heart. No significant accumulation of Mn was observed in tissues or organs. Long-term exposure to the heavy metal mixture mainly resulted in impairment to the circulatory system and reproductive system, along with liver dysfunction. To a certain degree, EGCG, TCD, or GSH could alleviate the joint toxicities of the heavy metal mixture and promote the excretion of Hg, Cd, Cr, Ni, and Cu, but with no significant effect on the discharge of Pb and Zn. The relationship between the transformation process of heavy metals and the mechanism of chemical antagonists in vivo remains to be further studied. In addition, our results indicate that the compensatory function of the kidneys, heart, lungs and spleen seems to be stronger than the liver and testis; in other words, the liver and testis seem to be more sensitive to multi-heavy metal-induced toxicities.

## Figures and Tables

**Figure 1 ijerph-17-01451-f001:**
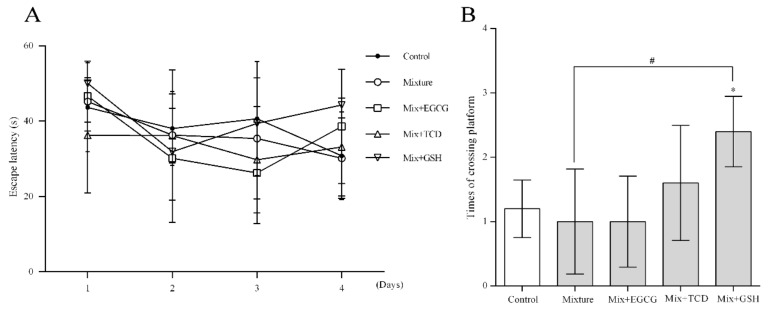
Effects of exposure to a heavy metal mixture on spatial learning and memory function, with or without the addition of EGCG, TCD, or GSH. The Morris water test was performed immediately after six months of treatment. Bars represented the times of crossing platform; (**A**) escape latency results of SD rats; (**B**) platform crossing times of SD rats; *n* = 6 rats/each group; Results were presented as means ± SD; Two-way ANOVA analysis was used in (**A**); One-way ANOVA analysis with LSD-t test was performed in (**B**). * *p* < 0.05, vs. the control group (Mixture, Mix + EGCG, Mix + TCD, Mix + GSH compared with the control group); ^#^
*p* < 0.05, vs. the Mixture group (Mix + EGCG, Mix + TCD, Mix + GSH compared with the Mixture group); Mix, Mixture.

**Figure 2 ijerph-17-01451-f002:**
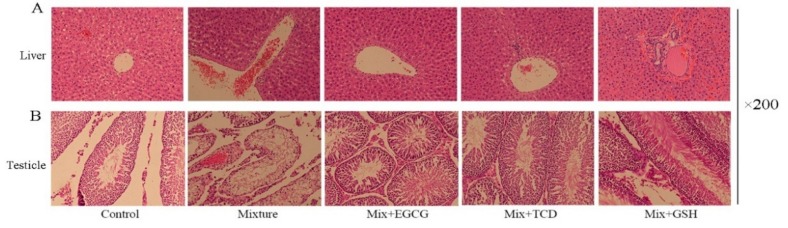
Effects of exposure to a heavy metal mixture on histomorphology of liver and testicle with or without the addition of EGCG, TCD or GSH. The formalin-fixed tissues were stored at 4 °C and the histopathological examination lasted two months. Mix, Mixture.

**Figure 3 ijerph-17-01451-f003:**
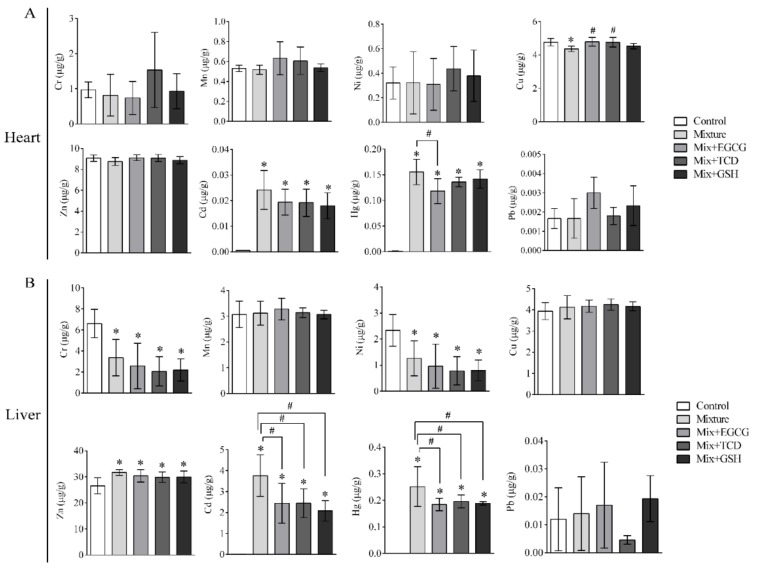
Contents of eight heavy metals in the heart and liver. ICP-MS was carried out after histopathological examination. Bars represented contents of eight heavy metals (ug) in per gram of organs; (**A**) Contents of eight heavy metals in the heart of SD rats; (**B**) Contents of eight heavy metals in the liver of SD rats; *n* = 6 rats/each group; Results were presented as means ± SD; One-way ANOVA and LSD-t test were showed; * *p* < 0.05, vs. the control group (Mixture, Mix + EGCG, Mix + TCD, Mix + GSH compared with the control group); ^#^
*p* < 0.05, vs. the Mixture group (Mix + EGCG, Mix + TCD, Mix + GSH compared with the Mixture group); Mix, Mixture.

**Figure 4 ijerph-17-01451-f004:**
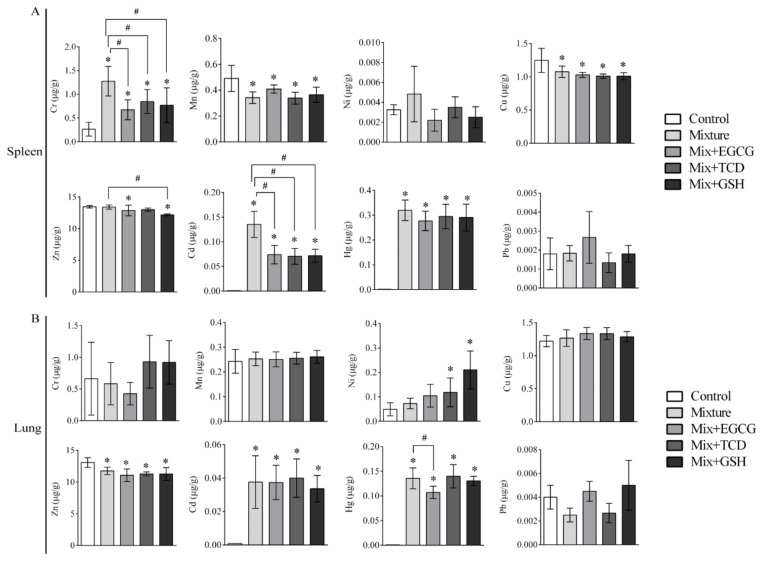
Contents of eight heavy metals in the spleen and lung of SD rats. Bars represented contents of eight heavy metals (ug) in per gram of organs; (**A**) Contents of eight heavy metals in the spleen of SD rats; (**B**) Contents of eight heavy metals in the lung of SD rats; *n* = 6 rats/each group; Results were presented as means ± SD; One-way ANOVA and LSD-t test were performed; * *p* < 0.05, vs. the control group (Mixture, Mix + EGCG, Mix + TCD, Mix + GSH compared with the control group); ^#^
*p* < 0.05, vs. the Mixture group (Mix + EGCG, Mix + TCD, Mix+GSH compared with the Mixture group); Mix, Mixture.

**Figure 5 ijerph-17-01451-f005:**
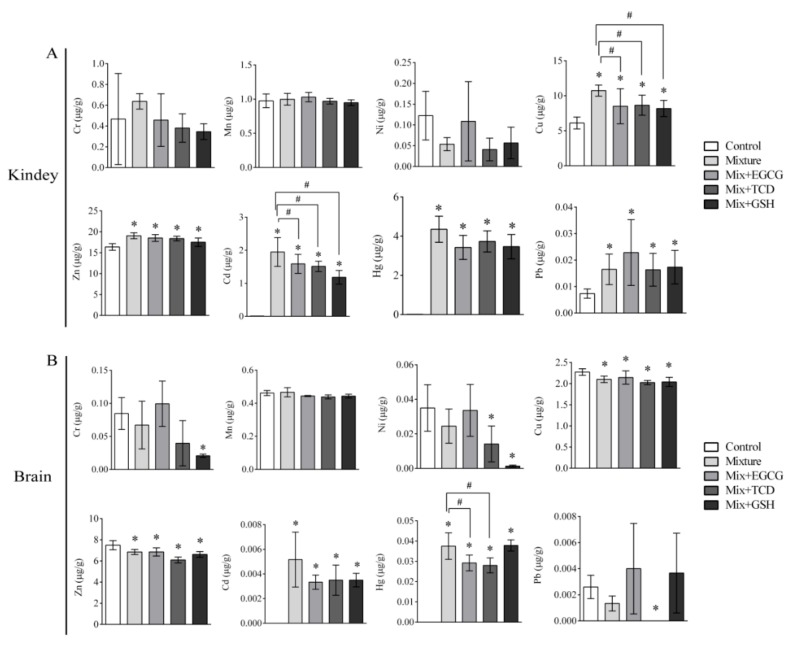
Contents of eight heavy metals in the kidney and brain of SD rats. Bars represented contents of eight heavy metals (ug) in per gram of organs; (**A**) Contents of eight heavy metals in the kidney of SD rats; (**B**) Contents of eight heavy metals in the brain of SD rats; *n* = 6 rats/each group; Results were presented as means ± SD; One-way ANOVA and LSD-t test were used. * *p* < 0.05, vs. the control group (Mixture, Mix + EGCG, Mix + TCD, Mix + GSH compared with the control group); ^#^
*p* < 0.05, vs. the Mixture group (Mix + EGCG, Mix + TCD, Mix + GSH compared with the Mixture group); Mix, Mixture.

**Figure 6 ijerph-17-01451-f006:**
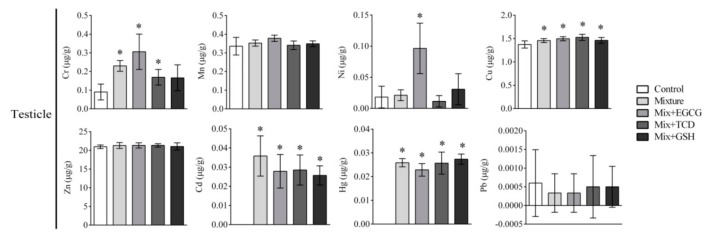
Contents of eight heavy metals in the testicle of SD rats. Bars represented contents of eight heavy metals (ug) in per gram of organs; *n* = 6 rats/each group; Results were presented as means ± SD; One-way ANOVA and LSD-t test were used. * *p* < 0.05, vs. the control group (Mixture, Mix + EGCG, Mix + TCD, Mix + GSH compared with the control group); Mix, Mixture.

**Table 1 ijerph-17-01451-t001:** The contents and proportions of different heavy metals composed in the mixture.

Chemical	Content (mg)	Mass Ratio of Metal Element
K_2_Cr_2_O_7_	8.547	41.345
MnCl_2_•4H_2_O	45.609	173.264
NiCl_2_•6H_2_O	8.893	30.054
CuSO_4_•5H_2_O	89.816	312.824
ZnSO_4_•7H_2_O	344.187	1070.838
CdCl_2_•2.5H_2_O	1.972	13.281
CH_3_ClHg	0.091	1.000
(CH_3_COO)_2_Pb_3_H_2_O	0.884	6.611
Total	500.001	

Morris Water Maze Test.

**Table 2 ijerph-17-01451-t002:** Effects of the heavy metal mixture on the organ weights with or without the addition of chemical antagonists in SD rats (*n* = 30).

(g)	Group					F	*p*
	Control	Mixture	Mix + EGCG	Mix + TCD	Mix + GSH		
Heart	1.68 ± 0.11	1.66 ± 0.12	1.54 ± 0.27	1.70 ± 0.13	1.72 ± 0.24	0.984	0.430
Liver	12.20 ± 0.98	12.58 ± 1.69	12.41 ± 1.48	12.88 ± 1.66	13.39 ± 1.82	0.647	0.633
Spleen	0.86 ± 0.12	0.77 ± 0.13	0.75 ± 0.11	0.81 ± 0.13	0.89 ± 0.07	1.938	0.128
Lung	1.67 ± 0.27	2.21 ± 0.52	1.75 ± 0.26	1.83 ± 0.48	1.77 ± 0.41	1.958	0.125
Kidney	3.33 ± 0.33	3.28 ± 0.29	3.27 ± 0.16	3.51 ± 0.28	3.52 ± 0.31	1.531	0.217
Brain	2.11 ± 0.10	2.09 ± 0.11	2.06 ± 0.05	2.09 ± 0.18	2.10 ± 0.08	0.136	0.968
Testicle	3.39 ± 0.17	3.16 ± 0.15 *	2.96 ± 0.21 *	3.18 ± 0.21 *	3.16 ± 0.19 *	3.923	0.011

Results were presented as means ± SD; One-way ANOVA and LSD-t test were showed; * *p* < 0.05, vs. the control group (Mixture, Mix + EGCG, Mix + TCD, Mix+ GSH compared with the control group); Mix, Mixture.

**Table 3 ijerph-17-01451-t003:** Effects of the heavy metal mixture on the organ/body weight coefficients with or without the addition of chemical antagonists in SD rats (*n* = 30).

	Group					F	*p*
	Control	Mixture	Mix + EGCG	Mix + TCD	Mix + GSH		
Heart	0.32 ± 0.02	0.32 ± 0.03	0.29 ± 0.05	0.30 ± 0.02	0.30 ± 0.04	0.983	0.431
Liver	2.29 ± 0.18	2.39 ± 0.38	2.31 ± 0.17	2.28 ± 0.14	2.29 ± 0.14	0.327	0.858
Spleen	0.16 ± 0.02	0.15 ± 0.02	0.14 ± 0.03	0.14 ± 0.02	0.15 ± 0.02	1.167	0.344
Lung	0.31 ± 0.04	0.42 ± 0.10 *	0.32 ± 0.05	0.33 ± 0.08	0.30 ± 0.07	3.025	0.032
Kidney	0.63 ± 0.09	0.62 ± 0.06	0.61 ± 0.03	0.63 ± 0.04	0.60 ± 0.03	0.295	0.879
Brain	0.39 ± 0.02	0.40 ± 0.03	0.39 ± 0.03	0.37 ± 0.04	0.36 ± 0.03	1.752	0.163
Testicle	0.64 ± 0.02	0.60 ± 0.03	0.55 ± 0.07 *	0.57 ± 0.05 *	0.54 ± 0.06 *	3.949	0.011

Results were presented as means ± SD; One-way ANOVA and LSD-t test were used; * *p* < 0.05, vs. the control group (Mixture, Mix + EGCG, Mix + TCD, Mix + GSH compared with the control group); Mix, Mixture.

**Table 4 ijerph-17-01451-t004:** Effects of the heavy metal mixture on the hematological parameters with or without the addition of chemical antagonists (*n* = 30).

Parameters	Group					F	*p*
	Control	Mixture	Mix + EGCG	Mix + TCD	Mix + GSH		
White Blood Cell							
WBC(×10^9^/L)	4.47 ± 0.93	3.27 ± 0.57	3.63 ± 0.91	3.20 ± 0.82	2.23 ± 0.47	3.362	0.055
NE%	31.57 ± 10.74	35.20 ± 9.10	27.17 ± 1.96	29.67 ± 2.60	30.40 ± 3.22	0.592	0.676
LY%	65.73 ± 9.90	61.46 ± 8.63	69.57 ± 3.41	65.73 ± 3.14	63.10 ± 4.72	1.176	0.353
MO%	1.07 ± 1.16	3.53 ± 1.57	1.43 ± 1.25	2.43 ± 1.16	3.03 ± 0.15	2.421	0.117
EO%	1.63 ± 0.35	2.80 ± 2.21	1.83 ± 0.49	1.60 ± 0.26	1.40 ± 0.61	0.800	0.552
Red Blood Cell							
RBC(×10^12^/L)	7.08 ± 0.35	7.16 ± 0.50	7.22 ± 0.37	7.12 ± 0.26	7.15 ± 0.40	0.093	0.984
HGB(g/L)	138.00 ± 6.32	137.17 ± 6.40	137.60 ± 5.50	136.83 ± 5.52	128.50 ± 17.07	1.069	0.393
HCT	0.40 ± 0.02	0.42 ± 0.03	0.41 ± 0.02	0.41 ± 0.03	0.39 ± 0.06	0.411	0.799
MCV(fl)	56.67 ± 2.45	57.50 ± 1.54	56.30 ± 1.27	57.28 ± 1.63	57.60 ± 1.58	0.417	0.795
MCH(pg)	19.78 ± 0.66	19.15 ± 0.55	19.10 ± 0.37	19.48 ± 0.67	19.40 ± 0.44	1.399	0.264
MCHC(g/L)	343.33 ± 8.16	333.17 ± 8.26	339.40 ± 6.11	340.00 ± 12.33	337.17 ± 9.24	1.011	0.421
RDW-CV(%)	14.62 ± 1.09	15.58 ± 0.97 *	14.68 ± 0.64 ^#^	14.73 ± 0.69	13.97 ± 0.38 ^#^	3.107	0.034
Platelet							
PLT(×10^9^/L)	881.33 ± 139.71	823.83 ± 73.52	777.60 ± 73.43	833.17 ± 40.09	849.67 ± 51.80	1.441	0.251
PDW-CV(%)	14.67 ± 0.05	14.65 ± 0.05	14.70 ± 0.10	14.70 ± 0.06	14.67 ± 0.05	0.675	0.616
MPV(fl)	5.78 ± 0.08	6.00 ± 0.14 *	5.94 ± 0.17 *	6.05 ± 0.08 *	5.97 ± 0.12 *	4.195	0.010
PCT	0.51 ± 0.08	0.53 ± 0.04	0.46 ± 0.04	0.52 ± 0.04	0.51 ± 0.03	1.487	0.237

Results were presented as means ± SD; One-way ANOVA and LSD-t test were included; * *p* < 0.05, vs. the control group (Mixture, Mix + EGCG, Mix + TCD, Mix + GSH compared with the control group); ^#^
*p* < 0.05, vs. the Mixture group (Mix + EGCG, Mix + TCD, Mix + GSH compared with the Mixture group).

**Table 5 ijerph-17-01451-t005:** Effects of the heavy metal mixture on serum biochemical indices with or without the addition of chemical antagonists (*n* = 30).

Parameters	Group					F	*p*
	Control	Mixture	Mix + EGCG	Mix + TCD	Mix + GSH		
TG(mmol/L)	0.62 ± 0.07	0.76 ± 0.45	0.77 ± 0.19	0.68 ± 0.39	0.68 ± 0.15	0.265	0.897
CHOL(mmol/L)	1.29 ± 0.30	1.39 ± 0.56	1.37 ± 0.28	1.29 ± 0.39	1.14 ± 0.18	0.417	0.794
TBIL(μmol/L)	2.85 ± 0.57	3.73 ± 0.84 *	2.37 ± 0.50 ^#^	2.72 ± 0.54 ^#^	2.50 ± 0.39 ^#^	3.667	0.021
DBIL(μmol/L)	0.83 ± 0.39	1.17 ± 0.47	0.57 ± 0.06	0.90 ± 0.31	0.75 ± 0.26	1.911	0.148
IBIL(μmol/L)	1.90 ± 0.36	2.57 ± 0.59 *	1.80 ± 0.56 ^#^	1.82 ± 0.44 ^#^	1.75 ± 0.18 ^#^	3.337	0.031
TP(g/L)	53.80 ± 2.80	56.17 ± 3.88	57.48 ± 2.62	53.98 ± 3.29	54.07 ± 3.00	1.469	0.243
ALB(g/L)	27.30 ± 1.17	29.03 ± 1.00	29.44 ± 1.22	27.90 ± 2.03	28.17 ± 1.78	1.620	0.202
GLOB(g/L)	26.50 ± 1.38	27.13 ± 2.94	28.04 ± 1.46	26.08 ± 1.46	25.90 ± 1.41	1.208	0.333
A/G(ALB/GLOB)	1.03 ± 0.05	1.08 ± 0.08	1.05 ± 0.03	1.07 ± 0.05	1.09 ± 0.04	1.033	0.411
ALT(U/L)	41.50 ± 7.74	45.67 ± 4.27	51.20 ± 6.67	42.17 ± 4.26	45.33 ± 9.50	1.705	0.182
AST(U/L)	145.83 ± 35.10	104.50 ± 30.46	146.80 ± 49.16	139.83 ± 17.17	129.33 ± 27.50	1.673	0.189
ALP(U/L)	83.17 ± 13.66	129.67 ± 22.49 *	119.40 ± 23.33 *	100.33 ± 8.98 ^#^	96.67 ± 22.18 ^#^	5.710	0.002
UREA(mmol/L)	6.74 ± 1.11	6.35 ± 0.62	6.80 ± 0.62	5.90 ± 0.82	6.17 ± 0.66	1.310	0.295
CREA(μmol/L)	35.07 ± 3.58	34.32 ± 3.17	33.64 ± 2.32	31.33 ± 1.13	32.08 ± 3.69	1.642	0.196
UR/Cr	0.19 ± 0.02	0.19 ± 0.02	0.20 ± 0.02	0.19 ± 0.03	0.20 ± 0.03	0.495	0.739
URCA(μmol/L)	81.83 ± 12.19	83.50 ± 17.75	90.60 ± 13.28	77.00 ± 15.77	78.00 ± 18.24	0.636	0.642

Results were presented as means ± SD; One-way ANOVA and LSD-t test were performed; * *p* < 0.05, vs. the control group (Mixture, Mix + EGCG, Mix + TCD, Mix + GSH compared with the control group); ^#^
*p* < 0.05, vs. the Mixture group (Mix + EGCG, Mix + TCD, Mix + GSH compared with the Mixture group).

**Table 6 ijerph-17-01451-t006:** The contents of eight heavy metals in the serum detected by ICP-MS (*n* = 30).

Element (µg/L)	Group					F	*p*
	Control	Mixture	Mix + EGCG	Mix + TCD	Mix + GSH		
^52^Cr	7.164 ± 0.624	9.271 ± 1.171 *	7.503 ± 0.607	8.197 ± 1.127	8.464 ± 1.441 *	3.129	0.035
^55^Mn	5.829 ± 0.974	9.123 ± 4.492	6.695 ± 1.409	6.374 ± 1.673	7.671 ± 3.327	1.267	0.311
^60^Ni	0.150 ± 0.245	3.012 ± 1.496 *	0.230 ± 0.124 ^#^	0.305 ± 0.264 ^#^	0.426 ± 0.294 ^#^	11.251	<0.001
^63^Cu	1056.750 ± 104.982	950.482 ± 107.573	964.510 ± 116.376	1010.820 ± 110.527	1122.620 ± 103.817	1.034	0.411
^66^Zn	1092.200 ± 125.269	1062.810 ± 125.269	1104.850 ± 100.431	1126.280 ± 162.298	1126.970 ± 137.700	0.252	0.905
^111^Cd	0.112 ± 0.096	0.826 ± 1.277	0.320 ± 0.191	0.253 ± 0.117	0.308 ± 0.119	1.198	0.338
^202^Hg	0.000 ± 0.000	9.712 ± 1.059 *	8.847 ± 1.356 *	10.476 ± 1.315 *	10.350 ± 0.968 *	111.310	<0.001
^208^Pb	1.422 ± 0.402	2.164 ± 1.106	1.710 ± 0.824	1.911 ± 0.496	2.126 ± 1.056	0.813	0.530

Results were presented as means ± SD; One-way ANOVA and LSD-t test were performed; * *p* < 0.05, vs. the control group (Mixture, Mix + EGCG, Mix + TCD, Mix+GSH compared with the control group); ^#^
*p* < 0.05, vs. the Mixture group (Mix + EGCG, Mix + TCD, Mix + GSH compared with the Mixture group); Mix, Mixture.

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
