# Peer review of "Joint Toxicity of a Multi-Heavy Metal Mixture and Chemoprevention in Sprague Dawley Rats"

_ijerph, 2020, doi:10.3390/ijerph17041451_

Round 1

Reviewer 1 Report

This work reported the joint toxicity of a multi-heavy metal mixture and chemoprevention in SD rats. This study is of great significance for environmental composite exposures. However, there are some specific comments as follows, which made my opinion that minor revisions will be needed when it come to publication in this journal.

Line 113-114: What is the basis for determining the exposure concentrations? Are the dosages of heavy metals, EGCG, TCD and GSH supported by references? The authors need to provide experimental or reference evidence. Line 178: Why was platform crossing time in the Mix+GSH group higher than that in control group?

    3. Line 254: Only the results of Morris water maze test could not show that exposure with multi-heavy metals induced no significant neurotoxicity. Did the authors tried otherl experiments?

Author Response

Thanks a lot for your good suggestions. We have answered the comments point-by-point. Please see the attachment.

Reviewer 2 Report

The manuscript presents the interesting results of the in vivo study of the phenotipic effects of the mixture of heavy metals and the chemopreventice effects of the three natural compounds, one xenobiotic and two endobiotic antioxidants. Although underlying assumption of the chemopreventive effects against heavy metal toxicities by the selected antioxidants is not novel, the manuscript provides many experimental results which are quite well presented and may be of interest to the scientists. In that regard, the manuscript may be recommended for the publication. However, it is strongly recommended to the authors to take into account the following points prior to the manuscript publication:

There may be some chemopreventive effects of the studied antioxidants on the rats as it is stated in the Conclusions section. Accordingly, the last two sentences in the Abstract should be reformulated. The last sentence in the Abstract has not been demonstrated enough in the manuscript. In the Introduction, the underlying assumption in the lines 59-60 has already been justified and accordingly the sentence should be reformulated and some references should be provided. In the line 62 „protons“ should be changed to „hydrogen atoms“ and in the line 65, „hydroxyphenol“ should be corrected to „phenolic OH groups“ . The sentence in the lines 67-68 is not clear enough – EGCG reduces some metal ions and transfers itself into prooxidative quinone forms which may be under certain conditions pro-oxidative and detrimental and it is suggested to reformulate this sentence. The objective of the study as it is described is not in accordance with the results provided. For example, no mechanisms have been investigated in the presented study. In the Material and methods the in vivo experiments should be much better described in terms of number of sacrificed animals in the experiments. It is also not clear enough at which time each of experiment was conducted. The latter should also be indicated in the Figure captions. It is not clear which rats are control group. Were there groups treated only by EGCG, TCD and GSH? If possible, it would be of great interpretation and discussion value to conduct multivariate statistical analysis on at least subset of data (eg those from Figures 3-5 and Table 6) like the principal component analysis.

Author Response

Response to reviewer comments for MS NO.: ijerph-692293

Dear reviewer,

Thanks for your constructive criticism and time in reviewing this manuscript (MS NO.: ijerph-692293). We have already modified the manuscript point to point according to the suggestions from you. Below is the response to the comments.

There may be some chemopreventive effects of the studied antioxidants on the rats as it is stated in the Conclusions section. Accordingly, the last two sentences in the Abstract should be reformulated. The last sentence in the Abstract has not been demonstrated enough in the manuscript. In the Introduction, the underlying assumption in the lines 59-60 has already been justified and accordingly the sentence should be reformulated and some references should be provided.

Response: Thanks for your good suggestion. We have corrected the last two sentences in the Abstract. Previous lines 59-60 “In light of these similarities in the toxic molecular mechanisms, we speculated that antioxidative chemicals might be potential agents for preventing multi-heavy-metal induced toxicities” have been changed to “These similarities contribute to searching for effective antioxidative chemicals for preventing multi-heavy-metal induced toxicities”. As we have described these similarities in lines 56-59 with relative references, this sentence serves as a summary sentence and introduces the following three chemical antagonists with more details. Therefore, we do not add references in this sentence. 

In the line 62, “protons” should be changed to “hydrogen atoms” and in the line 65, “hydroxyphenol” should be corrected to “phenolic OH groups”. 

Response: We have already changed “protons” to “hydrogen atoms”, and “hydroxyphenol” have been corrected to “phenolic OH groups”.

The sentence in the lines 67-68 is not clear enough – EGCG reduces some metal ions and transfers itself into prooxidative quinone forms which may be under certain conditions pro-oxidative and detrimental and it is suggested to reformulate this sentence.

 Response: The sentence in the previous lines 67-68 “Simultaneously, EGCG catalyzes high-valent metal ions into low-valent metal ions and oxidizes itself to quinone or other derivatives, thus reducing the absorption or toxicity of heavy-metal ions” has been corrected to “Therefore, EGCG has a potential protective effect against heavy metal-induced toxicity”.

The objective of the study as it is described is not in accordance with the results provided. For example, no mechanisms have been investigated in the presented study. 

Response: Thanks for your constructive suggestion. The objective of the study which is referred to “mechanisms” occurs in the last two sentence in the Introduction section “The objective of this study was to investigate the joint toxicity and underlying mechanisms following chronic exposure to a multi-heavy metal mixture through in vivo experimentation”. We have changed the sentence to “The objective of this study was to investigate the joint toxicity following chronic exposure to a multi-heavy metal mixture through in vivo experimentation”, thus according with our current study. 

In the Material and methods the in vivo experiments should be much better described in terms of number of sacrificed animals in the experiments. It is also not clear enough at which time each of experiment was conducted. The latter should also be indicated in the Figure captions. It is not clear which rats are control group.

 Response: I have added the number of animals (n=30) in the Material and Methods section. Chronological sequence of each experiment has been showed in the article in the lines 148 and 154-158. I also added the times of each experiment in the figure captions in lines 191-192, 250-251 and 257-258. The control was the control group. We made two comparisons in this paper. The first is among the five groups: four mixture-treated groups (mixture, Mix+EGCG, Mix+TCD, Mix+GSH) that were compared with the control group, the aim of which is to demonstrate the combined toxicity of multi-heavy metals. The second comparison is among the four mixture-treated groups, that is the chemical antagonists treated groups(Mix+EGCG, Mix+TCD, Mix+GSH) which were compared with the mixture group, the aim of which is to verify the protective effects of EGCG, TCD,GSH against the detrimental effects induced by multi-heavy metals. 

Were there groups treated only by EGCG, TCD and GSH? If possible, it would be of great interpretation and discussion value to conduct multivariate statistical analysis on at least subset of data (eg those from Figures 3-5 and Table 6) like the principal component analysis.

Response: In this experiment, we do conducted groups treated only by EGCG, TCD and GSH and the results showed no significance with the control group and do not cause harmful results. The final doseage 200 mg/kg of cytoprotectants (EGCG, GSH, TCD) was referred to research articles like (Chen J, Du L, et al. Epigallocatechin-3-gallate attenuates cadmium-induced chronic renal injury and fibrosis. Food Chem Toxicol. 2016, 96:70-8.) (Abdel-Salam OM, Youness ER, et al. Citric acid effects on brain and liver oxidative stress in lipopolysaccharide-treated mice. J Med Food. 2014, 17(5):588-98.) (Wang W. The study of effects of GSH, Vit_C and DMPS on the toxicity induced by cadmium and their mechanisms (D). 2004. China Medical University, China). Therefore, relative results were not showed in this article. Since sample size is relatively small, just 6 in each group, performing multivariate statistical analysis may cause large errors. Therefore, we just give an objective description of the amount of eight heavy metals in organs and serum.

I hope we have answered all questions point to point. If you have any questions, please let me know.

Thank you once again for your consideration in publishing this manuscript.

Yours Sincerely

Jinshun Zhao

M.D., Ph. D, Professor, Dean of the Department of Preventive Medicine,

Medical School of Ningbo University, 818 Fenghua Road, Jiangbei District, Ningbo 315211, Zhejiang Province, P.R. China

Tel: +86 0574-8760-9591

Fax: +86 0574-8760-8638

E-mail: zhaojinshun@nbu.edu.cn

Reviewer 3 Report

The idea to study simultaneous exposure seems to be very novel. The choice of exposure levels/doses seems unrealistic however this study can be a foundation of future refined studies in this space.

Measuring the amount of heavy metals in the faeces would have provided more information of the actual exposure quantities to the organs - amount of heavy metals absorbed in the stomach. An attempt to balance the dose administered to the quantities measured in the organs and faeces would have helped in understanding the exposure more. 

Author Response

Thanks for your suggestions, we have answer your comments point-by-point. Please see the  attachment.

Round 2

Reviewer 2 Report

Fine/minor spell and small/capital letter checks are required.

Author Response

Thanks for your nice seggestions. We have read our article carefully again and made some corrections according to your comments.